# Phonon Pumping by Modulating the Ultrastrong Vacuum

Fabrizio Minganti[1,2*], Alberto Mercurio[1,2,3†], Fabio Mauceri[3], Marco Scigliuzzo[2,4], Salvatore Savasta[3] and Vincenzo Savona[1,2]

**1** Laboratory of Theoretical Physics of Nanosystems (LTPN), Institute of Physics, Ecole Polytechnique Fédérale de Lausanne (EPFL), CH-1015 Lausanne, Switzerland
**2** Center for Quantum Science and Engineering, EPFL, CH-1015 Lausanne, Switzerland
**3** Dipartimento di Scienze Matematiche e Informatiche, Scienze Fisiche e Scienze della Terra, Università di Messina, I-98166 Messina, Italy
**4** Laboratory of Photonics and Quantum Measurements (LPQM), Institute of Physics, EPFL, CH-1015 Lausanne, Switzerland

⋆ fabrizio.minganti@gmail.com , † alberto.mercurio96@gmail.com

## Abstract

The vacuum (i.e., ground state) of a system in ultrastrong light-matter coupling contains particles that cannot be emitted without any dynamical perturbation, and thus called virtual. We propose a protocol for inducing and observing real mechanical excitations of a mirror enabled by the virtual photons in the ground state of a tripartite system, where a resonant optical cavity is ultrastrongly coupled to a two-level system (qubit) and, at the same time, optomechanically coupled to a mechanical resonator. Real phonons are coherently emitted when the frequency of the two-level system is modulated at a frequency comparable to that of the mechanical resonator and, therefore much lower than the optical frequency. We demonstrate that this hybrid effect is a direct consequence of the virtual photon population in the ground state. Within a classical physics analogy, attaching a weight to a spring only changes its resting position, whereas dynamically modulating the weight makes the system oscillate. In our case, however, the weight is the vacuum itself. We propose and accurately characterize a hybrid superconducting-optomechanical setup based on available state-of-the-art technology, where this effect can be experimentally observed.

# 1   Introduction

The ultrastrong coupling (USC) regime between light and matter occurs when the coupling connecting the two is a significant fraction of their quantized resonance frequencies [1]. In the USC regime of the quantum Rabi model, counter-rotating coupling terms, which do not conserve the number of particles, lead to an entangled ground state with nonzero particles [2, 3]. Similar to zero-point energy, these particles cannot be converted into real excitations that could be emitted or detected, unless the system is dynamically perturbed over a timescale comparable to the period of optical oscillations [2, 4–6]. In this sense, the *vacuum* (i.e., ground state) of a USC system contains *virtual* particles. USC regime has been achieved in various platforms like superconducting circuits, intersubband polaritons, and magnonic systems [7–16].

In an optomechanical system, the radiation pressure of the electromagnetic field displaces one of the mirrors of the cavity. This displacement, in turn, modulates the cavity's resonance frequency [17]. Optomechanical coupling has found numerous applications [17, 18], such as ground-state cooling of the mechanical mode [19–21], generation of nonclassical states [22], and macroscopic entanglement [23, 24]. The vacuum fluctuations of the quantum electromagnetic field sum to determine the total energy of a system in the ground state, leading to, e.g., the Casimir effect [25–27]. Dynamical perturbations of the mirror can convert virtual photons into real photons, resulting in the dynamical Casimir effect [28], which has been quantum simulated using a superconducting circuit architecture [29, 30]. There is an increasing interest in achieving larger optomechanical couplings [31–33], enabling the possibility of directly observing the dynamical Casimir effect and other peculiar effects arising from it [34–36]. Systems combining a USC part and an optomechanical one, involving virtual and optomechanical transitions, have been recently proposed [37, 38].

Here, we propose a novel effect, where virtual photons in a hybrid USC-optomechanical system can give rise to real mechanical excitations. To do that, we periodically modulate the vacuum (i.e., ground state) energy using the features of USC coupling. This novel effect bears close resemblance to the Casimir effect, where the space modulation of energy density

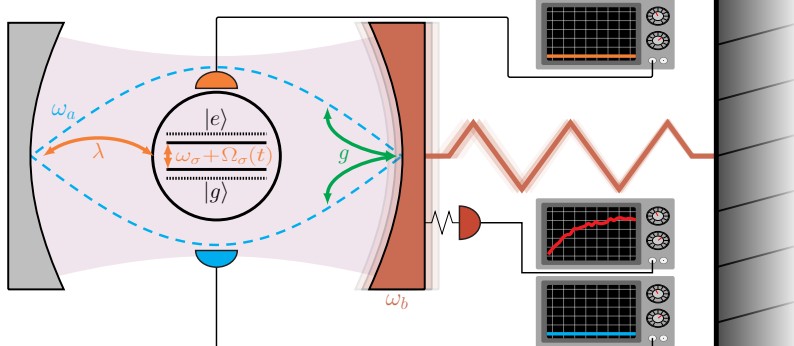

Figure 1: Schematic depiction of the system. A cavity at frequency $\omega_a$ and a qubit of bare frequency $\omega_\sigma$, are in USC with coupling strength $\lambda$. The cavity also interacts with a mirror, whose vibration frequency is $\omega_b$, through an optomechanical coupling of intensity $g$. The frequency of the qubit is adiabatically modulated through $\Omega_\sigma(t)$, and the virtual photon population oscillates in time. This causes the mirror to oscillate. Collecting the emission of both the USC systems and of the vibrating mirror, only the latter will produce a signal.

between the two sides of a mirror is what ultimately induces the Casimir force. Here, the time modulation of the energy results in an effective drive of the mirror through the USC vacuum, in stark contrast to a standard optomechanical drive, where the cavity is driven far from its ground state and consequently gets populated by a large number of real photons.

The system we propose to realize such an effect is a tripartite USC-optomechanical system. The architecture, depicted in Fig. 1, includes a cavity in USC with a two-level system (qubit). The cavity is additionally an optomechanical system. The ground state of the USC system contains virtual particle, whose presence influences the mechanical degrees of freedom. The frequency of the qubit is periodically modulated, with a period much longer than that characterizing the oscillations of the USC components, but coinciding with that of the mechanical oscillation. While the USC subsystem adiabatically remains in the ground state, which does not emit photons into the environment, the number of *virtual* ground-state photons determines the ground state energy, and its modulation create the *real* (i.e., detectable) mechanical oscillations of the mirror. We propose an experimental protocol to observe this virtual-to-real transduction in advanced hybrid superconducting optomechanical systems.

## 2   Model

Let $\hat{a}$ ($\hat{a}^\dagger$) be the annihilation (creation) operators of the cavity mode, $\hat{b}$ ($\hat{b}^\dagger$) of the mirror vibration mode, and $\hat{\sigma}_-$ and $\hat{\sigma}_+$ the Pauli operators associated with the qubit. As detailed in the Appendix D, the system is described by the Hamiltonian ($\hbar = 1$):

$$\hat{H}(t) = \hat{H}_\mathrm{R} + \hat{H}_\mathrm{opt} + \hat{H}_\mathrm{M}(t)$$
$$\hat{H}_\mathrm{R} = \omega_a \hat{a}^\dagger \hat{a} + \omega_\sigma \hat{\sigma}_+ \hat{\sigma}_- + \lambda(\hat{a} + \hat{a}^\dagger)(\hat{\sigma}_- + \hat{\sigma}_+),$$
$$\hat{H}_\mathrm{opt} = \omega_b \hat{b}^\dagger \hat{b} + \frac{g}{2}(\hat{a} + \hat{a}^\dagger)^2(\hat{b}^\dagger + \hat{b}). \tag{1}$$

$\hat{H}_\mathrm{R}$ is the Rabi Hamiltonian giving rise to the USC interaction, and $\hat{H}_\mathrm{opt}$ is the optomechanical coupling, up to a constant displacement of the phononic field. $\hat{H}_\mathrm{opt}$ is derived from first principles both in the case of an electromagnetic field coupled to a vibrating mirror [39] and

70  for circuital analogs [40]. It includes the rapidly rotating terms $(\hat{a}^2 + \hat{a}^{\dagger 2})(\hat{b}^{\dagger} + \hat{b})$ [39, 40]
71  which, as will emerge from our analysis, cannot be neglected in the present protocol [41]. We
72  assume a modulation of the qubit resonance frequency of the form

$$\hat{H}_{\mathrm{M}}(t) = \frac{1}{2}\Delta_{\omega}\left[1 + \cos(\omega_d t)\right]\hat{\sigma}_+\hat{\sigma}_- = \Omega_{\sigma}(t)\hat{\sigma}_+\hat{\sigma}_- . \tag{2}$$

73      The regime of interest is one where $g \ll \omega_d \simeq \omega_b \ll \omega_a \simeq \omega_{\sigma}$. In this regime, entangle-
74  ment between the mechanical motion and the USC subsystem is negligible, and the state of the
75  system can be factored as $|\Psi(t)\rangle \simeq |\psi(t)\rangle \otimes |\phi_b(t)\rangle$, where $|\psi(t)\rangle$ describes the USC state, and
76  $|\phi_b(t)\rangle$ is the one of the mirror. A further approximation, holding because $\omega_d \ll \omega_a \simeq \omega_{\sigma}$, is
77  that the USC subsystem adiabatically remains in its vacuum $|\psi(t)\rangle = |\psi_0(t)\rangle$, where $|\psi_0(t)\rangle$
78  is the ground state of $\hat{H}_{\mathrm{R}} + H_{\mathrm{M}}(t)$.
79      Under these approximations, the time-dependent Hamiltonian governing the motion of the
80  mirror is

$$\hat{H}_b(t) = \langle\psi_0(t)|\hat{H}_{\mathrm{opt}}|\psi_0(t)\rangle = \omega_b \hat{b}^{\dagger}\hat{b} + \frac{g}{2}\mathcal{N}(t)\left(\hat{b} + \hat{b}^{\dagger}\right) , \tag{3}$$

81  where $\mathcal{N}(t) \equiv \langle\psi_0(t)|2\hat{a}^{\dagger}\hat{a} + \hat{a}^2 + \hat{a}^{\dagger 2}|\psi_0(t)\rangle$ is the time-dependent radiation pressure, act-
82  ing as a drive on the mirror and generating real phonons (i.e., detectable).
83      Drawing a parallel with classical physics, where increasing the weight on a spring merely
84  alters its equilibrium state, $N(t)$ dynamically modifies the "weight" attached to the spring.
85  Interestingly, in our scenario, the weight is the vacuum itself.
86      The full system dynamics is then governed by the Hamiltonian $\hat{H}_{\mathrm{eff}}(t) = \hat{H}_R + \hat{H}_M(t) + \hat{H}_b(t)$.
87  Notice the importance of the counter-rotating terms $\hat{a}\hat{\sigma}_-$ and $\hat{a}^{\dagger}\hat{\sigma}_+$ in $\hat{H}_{\mathrm{R}}$: if they are ne-
88  glected, one wrongly predicts $\mathcal{N}(t) = 0$. This shows that the mirror oscillates only if the
89  ground state contains virtual photons.
90      The validity of these approximations is assessed in Fig. 2 by simulating the system dynam-
91  ics both under the full Hamiltonian $\hat{H}(t)$ in Eq. (1) and the effective Hamiltonian $\hat{H}_{\mathrm{eff}}(t)$. The
92  quantity $\mathcal{N}(t)$ is plotted in Fig. 2(a). The number of phonons is shown in Fig. 2(b). As the
93  full and the effective dynamics are in excellent agreement, we conclude that our interpretation
94  holds, and that the phonon number increases in time due to the radiation pressure originating
95  from $\mathcal{N}(t)$.

## 3   Open-system dynamics

97  Experimental devices are always subject to the influence of the environment, which has a finite
98  temperature and generally induces loss and dephasing. For the parameters we consider, the
99  finite temperature of the environment in, e.g., a dilution refrigerator ($T \approx 10$ mK) leads to
100 thermal noise in the phononic part ($n_{\mathrm{th}} \approx 200$) but not in the photonic one ($n_{\mathrm{th}} \approx 5 \times 10^{-9}$)
101 [1]. The open system dynamics, when assuming a Markovian environment, is governed by the
102 Lindblad master equation

$$\dot{\hat{\rho}} = -i\left[\hat{H}(t), \hat{\rho}\right] + (1 + n_{\mathrm{th}})\gamma_b \mathcal{D}\left[\hat{b}\right]\hat{\rho}$$
$$+ n_{\mathrm{th}}\gamma_b \mathcal{D}\left[\hat{b}^{\dagger}\right]\hat{\rho} + \gamma_{\mathrm{D}}\mathcal{D}\left[\hat{b}^{\dagger}\hat{b}\right]\hat{\rho}, \tag{4}$$

103 where $\hat{\rho}$ is the density matrix of the system and $\mathcal{D}[\hat{O}]\hat{\rho} = 1/2(2\hat{O}\hat{\rho}\hat{O}^{\dagger} - \hat{\rho}\hat{O}^{\dagger}\hat{O} - \hat{O}^{\dagger}\hat{O}\hat{\rho})$ is the
104 Lindblad dissipator. The phonon loss at rate is $\gamma_b(1 + n_{\mathrm{th}})$, the gain $\gamma_b n_{\mathrm{th}}$, and the dephasing

---

[1]Assuming a Bose-Einstein distribution induced by the reservoir, one gets $n_{\mathrm{th}} = [\exp(\hbar\omega_j/k_{\mathrm{B}}T) - 1]^{-1}$ for the chosen frequency. Nevertheless, it is known that qubit can experience non-equilibrium thermal population.

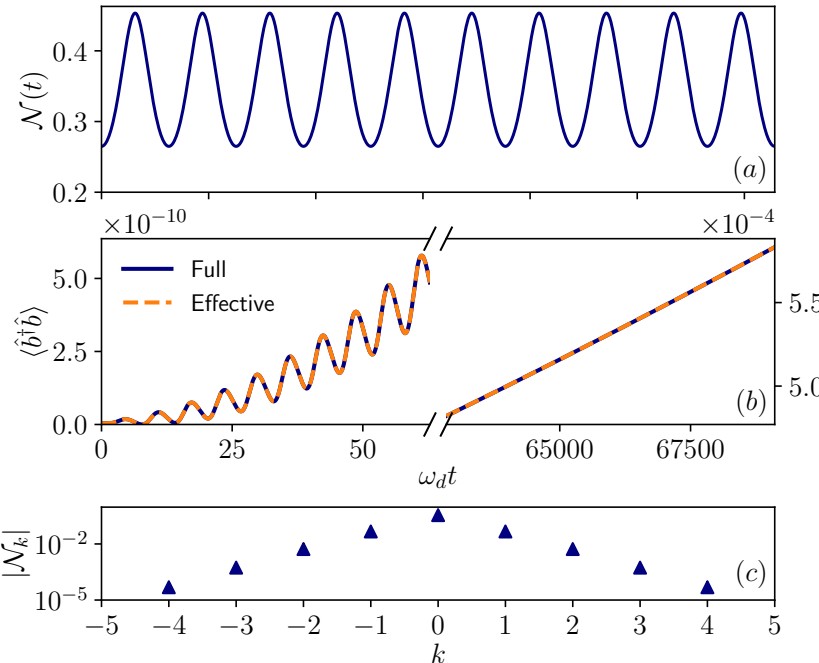

Figure 2: (a) $\mathcal{N}(t)$ in Eq. (3) obtained by simulating Eq. (1). (b) Number of phonons as a function of time according to the full Hamiltonian in Eq. (1) (blue solid line) and the effective Hamiltonian in Eq. (3) (orange dashed line). The two curves are in excellent agreement, validating the approximations in Eq. (3). (c) Fourier components $\mathcal{N}_\kappa$ of $\mathcal{N}(t)$. Parameters: $\omega_a = \omega_\sigma = 2\pi \times 4\text{GHz}$, $\omega_b = \omega_d = 2\pi \times 1\text{MHz}$, $\lambda = 0.5\omega_a$, and $g = 2\pi \times 15\text{Hz}$, comparable to Ref. [42]. The system is initialized in the ground state of $\hat{H}(t=0)$.

$\gamma_\text{D}$, with $n_\text{th}$ the thermal population [43]. As we have verified, the USC subsystem remains in its ground state, and thus dissipation processes are absent, despite the finite number of virtual photons. Indeed, when describing an open USC system, dissipation must result in the exchange of *real* excitations between the system and the environment, rather than virtual ones [6,44]. At $T = 0$ in particular, the system can only lose energy to the environment, through the emission of real photons. In an ideal setup, never detecting photons but observing the vibration of the mirror is thus the signature that virtual photons are generating a radiation pressure (see Fig. 1). We therefore do not include photon loss terms in Eq. (4). As for the mechanical part, all dissipators can be expressed in terms of the bare phonon operators $\hat{b}$ and $\hat{b}^\dagger$, as the excitations of the mechanical mode are real, and not virtual. Indeed, the ground state of the mechanical mode is almost empty ($\langle \hat{b}^\dagger \hat{b} \rangle < 10^{-12}$). Furthermore, the effective Hamiltonian in Eq. (3) has been numerically verified to be valid also in the presence of dissipation [41].

## 4   Main features of the model

As $\hat{H}_\text{M}(t)$ has period $2\pi/\omega_d$, we decompose $\mathcal{N}(t)$ in its Fourier components as

$$\mathcal{N}(t) = \sum_{k=-\infty}^{+\infty} \mathcal{N}_k \exp[i \, k \omega_d \, t].$$ 

(5)

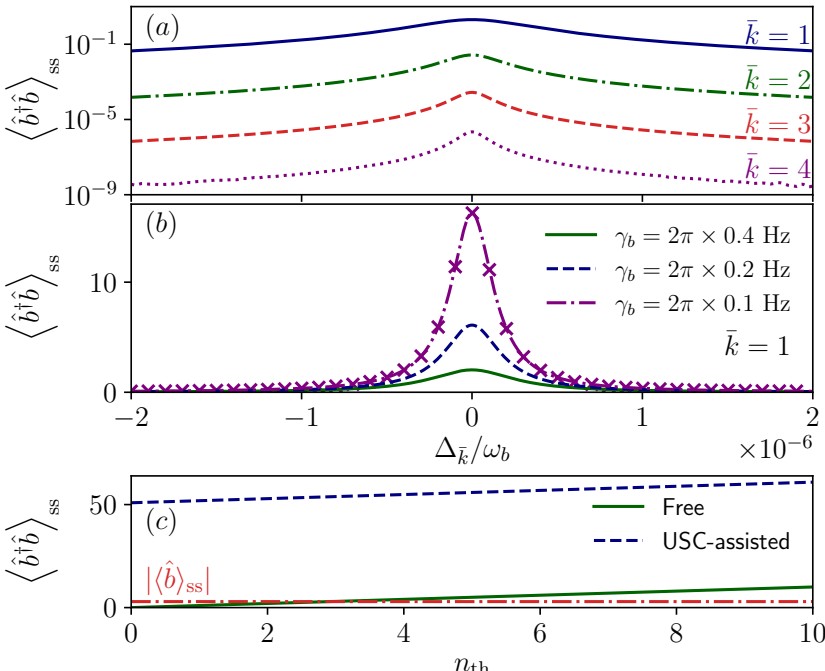

Figure 3: Steady state phonon population: (a) With different drive frequencies: $\omega_d \simeq \omega_b/\bar{k}$. The number of phonons follows the magnitude of the Fourier coefficients $\mathcal{N}_{\bar{k}}$, which are shown in Fig. 2(c); (b) Varying the detuning $\Delta_{\bar{k}}$ with $\bar{k} = 1$ and with three different values of the mechanical damping $\gamma_b$. The purple × points represent the analytical steady state population, following Eq. (6), which is perfectly in agreement with the numerical simulations; (c) As a function of the thermal population $n_{\text{th}}$, showing both the cases $\Delta_\omega = 0$ (green line) and $\Delta_\omega = 4$ GHz (blue line). The effect of $n_{\text{th}}$ is to linearly increase the steady state population as in Eq. (6). Parameters as in Fig. 2, and, if not specified, $\gamma_b = 2\pi \times 400$ mHz and $\gamma_D = 2\pi \times 200$ mHz. For this choice, the steady state is reached in a time $1/\gamma_b \approx 1$s.

The effective drive resonance condition then occurs for $\omega_d = \omega_b/\bar{k}$ with $\bar{k} > 0 \in \mathbb{N}$, as confirmed by the numerical simulation reported in Fig. 2(c). If we now assume to be close to resonance with the $\bar{k}$th component, so that the "pump-to-cavity detuning" $\Delta_{\bar{k}} \equiv \bar{k}\omega_d - \omega_b \simeq 0$, and passing in the frame rotating at $\omega_d$, we can discard fast rotating terms [2] and obtain [41]

$$\langle \hat{b}^\dagger \hat{b} \rangle_{\text{ss}} = \frac{\gamma + \gamma_D}{\gamma} \left| \langle \hat{b} \rangle_{\text{ss}} \right|^2 + n_{\text{th}}, \tag{6}$$

with $\langle \hat{b} \rangle_{\text{ss}} = (g\mathcal{N}_{\bar{k}})/[2\Delta_{\bar{k}} + i(\gamma + \gamma_D)]$.

In experimental implementations, the optomechanical coupling $g$ is a limiting factor in reaching large $\langle \hat{b}^\dagger \hat{b} \rangle_{\text{ss}}$. Choosing $\omega_d \approx \omega_b$ achieves the largest value of $\mathcal{N}_k$, thus enhancing the driving effect. Furthermore, the low loss rate in optomechanical systems, and the large values of $\lambda$ (and thus of $\mathcal{N}_k$) realized in superconducting circuit architectures [2], make the phenomenon detectable according to our estimates.

---

[2]The rotating wave approximation can only be performed at this stage. Performing it before, instead, would result in neglecting resonant, thereby important, terms.

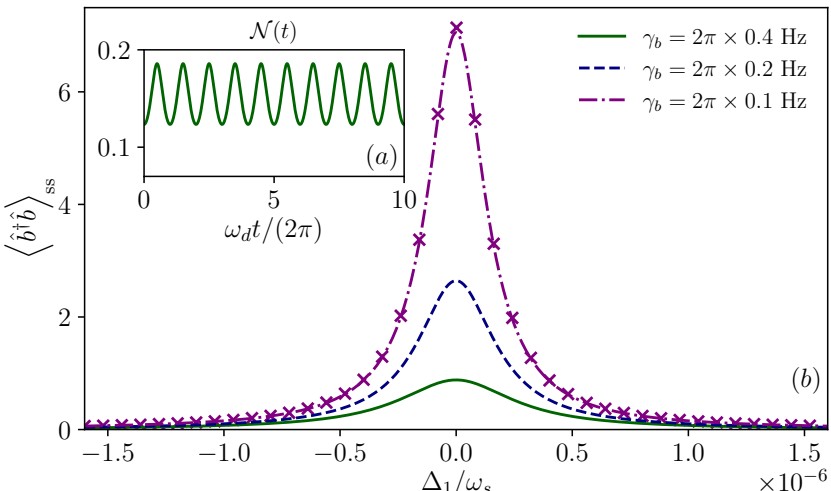

Figure 4: As in Fig. 3(b), the phonon population at the steady state, but when the USC part is described by two interacting harmonic resonators. The markers represent the analytical steady state population, obtained by generalizing Eq. (6). Inset: the time evolution of $\mathcal{N}(t)$. The used parameters are the same as Fig. 3, except for $g = 2\pi \times 30$ Hz, $\Delta_\omega = 2\pi \times 2$ GHz, and $\lambda = 0.3\omega_a$.

## 5 Results

Fig. 3 shows the creation of mechanical excitations by modulating the properties of the USC vacuum (i.e., ground state) in a dissipative environment. The mechanical part of the system reaches a periodic steady regime in a timescale of the order $1/\gamma_b$, the details depending on the specific choice of parameters. This Floquet steady state was numerically obtained by using the Arnoldi-Lindblad algorithm [45] for Eq. (4) using the approximation in Eq. (3). Fig. 3(a) shows the steady state population as a function of the frequency of the modulation $\omega_d$ at the resonance condition $\omega_d = \omega_b/\bar{k}$ and for different values of $\bar{k}$. The validity of Eq. (6), and the profound impact of the dissipation rate, is shown in Fig. 3(b), where we plot the steady-state population as a function of the frequency of the modulation $\omega_d$. Both the analytical prediction and the numerical result have a Lorentian profile, and they perfectly match (shown only for one curve). The impact of the thermal population $n_{\text{th}}$ on both the coherence and the total number of phonons is shown in Fig. 3(c). As predicted by Eq. (6), thermal phonons do not modify the coherent emission, but they result in a background phonon occupation that can be subtracted in the experimental analysis.

We have thus shown that virtual photons can pump mechanical vibrations.

### 5.1 Two-linear photonic cavities

Above we considered a two-level system in interaction with a linear cavity. Experimentally, two level systems are realized by means of large nonlinearities, which may prove difficult to realize in actual hybrid optomechanical architecturess. For this reason, here we demonstrate that the same phonon pumping by virtual photons can be obtained even if we assume that the USC part consists in two coupled harmonic cavities. This model is described by replacing the two-level systems with a bosonic operator ($\hat{\sigma}_- \to \hat{c}$ and $\hat{\sigma}_+ \to \hat{c}^\dagger$, where $\hat{c}$ is the bosonic field) [46, 47].

The same analysis reported in Fig. 3 is repeated for this linear model in Fig. 4. All the results lead to the same conclusion as in the nonlinear case, and are in agreement with the

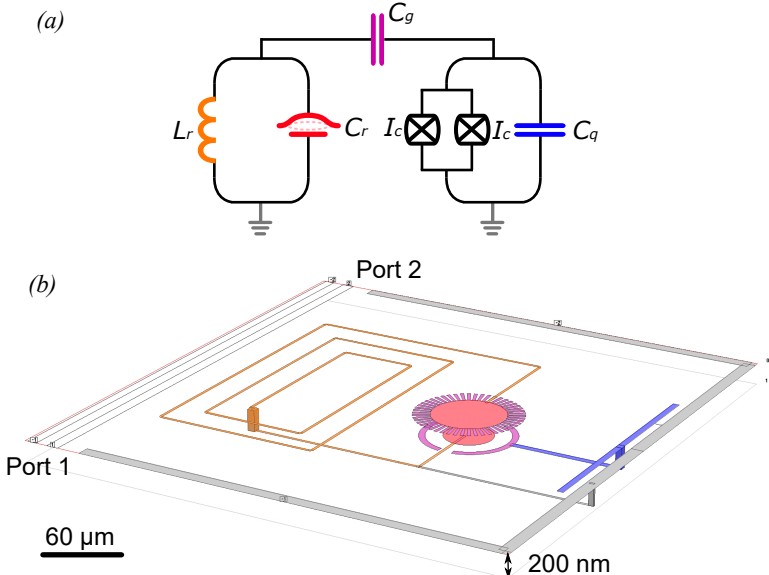

Figure 5: (a) Lumped element circuit of a resonator composed by a linear inductor $L_{\mathrm{r}}$ and a mechanical-compliant vacuum-gap capacitor ($C_{\mathrm{r}}$), capacitively coupled (though $C_g$) to a frequency tunable transmon qubit. The transmon is realized by a capacitor $C_{\mathrm{q}}$ in parallel to a SQUID (with Josephon junctions of identical critical current $I_{\mathrm{c}}$). (b) Design simulated in SONNET® of a 60 $\mu m$ mechanical drum with 200 $nm$ vacuum gap to the bottom electrode. The circuit parameters are extracted by the signal transmission between port 1 and port 2.

generalization of Eq. (6) to linear models. Considerations about the largest frequency shift that can be induced in current experimental systems lead to the conclusion that the maximal occupation of the phononic mode is smaller than in the nonlinear case.

## 6   Design and simulation of an experimental device

This model can be realized in superconducting circuit architectures. For instance, the qubit is implemented by a flux-tunable transmon capacitively coupled to a lumped element LC resonator, that is the cavity. The latter is formed by shunting an inductance and a mechanically compliant parallel plate capacitor (the vibrating mirror) [29, 48, 49]. The proposed schematics is shown in Fig. 5(a). We model the transmon as a bosonic cavity of initial frequency $\omega_\sigma$ characterized by a Kerr interaction of the form $\chi(\hat{c}^\dagger)^2\hat{c}^2$. A periodic modulation of the magnetic flux threaded in the transmon SQUID loop by an on-chip flux line results in $\hat{H}_{\mathrm{M}}(t) = \Delta_\omega \sin(\omega_d t)\hat{c}^\dagger\hat{c}$. Such a modulation would also change the coupling strength $\lambda(t) = \lambda(0)\sqrt{1 + \Delta_\omega \sin(\omega_d t)/\omega_\sigma}$. We provide a detailed derivation in the Appendix C.

Based on these target parameters we design the device shown in Fig. 5(b). By simulating the system with the SONNET® software, we obtain: $\omega_a = 2\pi \times 9.2$ GHz; $\omega_\sigma = 2\pi \times 9.2$ GHz for a 4 $nH$ lumped element inductor that is used to simulate the SQUID. This correspond to $C_{\mathrm{q}} + C_g = 75$ fF, i.e. $\chi = 2\pi \times e^2/2h(C_{\mathrm{q}} + C_g) = 2\pi \times 270$ MHz; $\lambda_0 = 0.26\omega_a$. From the drum diameter, we estimate $\omega_b = 2\pi \times 3.8$ MHz and the optomechanical coupling $g = 2\pi \times 15$ Hz; $\Delta_\omega = 2\pi \times 7$ GHz. For these parameters we have: $|\langle\hat{b}\rangle| \simeq 1.2$ and $\langle\hat{b}^\dagger\hat{b}\rangle \simeq 8.4 + n_{\mathrm{th}}$.

### 6.1 Phonon occupation readout

The readout of the phononic field can be performed as in Ref. [42]. One switches off the qubit modulation and pumps the cavity, with a signal red-detuned of $\omega_b$. Recording the microwave quadrature at frequency $\omega_a$ while the pump is active allows reconstructing the *coherent* part of the phononic field (i.e., the one produced only by the vacuum modulation). Following this protocol, signal-to-noise ratio of up to few percent can be detected averaging over many relaxation cycles [42]. State-of-the art experiments allow reaching the steady state in a dilution refrigerator with a base temperature of 10 mK, and thus a signal of $\langle \hat{b}^\dagger \hat{b} \rangle \simeq 8.4 > 10\% n_{\text{th}} \approx 56$ [43]. Notice that the very same cavity $\omega_a$ can be directly used to probe the phononic mode.

## 7 Conclusions

We have considered a USC system optomechanically coupled to a mechanical mirror. We demonstrate both numerically and semi-analytically how the presence of modulated virtual photons – i.e., photons that cannot be emitted into the environment – enables a *real* mechanical vibration on the mirror. We have demonstrated that this effect can be realized using current experimental platforms, and we show an explicit example of a hybrid superconducting circuit implementation.

The key features of this system are: (i) the mirror vibrates when the frequency of the modulation matches that of the phononic mode (or integer fractions of it); (ii) despite the fact that the mirror vibrates, and these vibrations can be detected, no photons are emitted by the USC subsystem (see the sketch in Fig. 1); (iii) although thermal population contributes to the total number of phonons, the only coherent contribution comes from the effective drive induced by the virtual photons.

The remarkable conclusion of our proposal is that virtual photons can drive real mechanical excitations. This phenomenon presented here bears clear similarities with the dynamical Casimir effect predicted for USC systems. The important difference is that, however, the external periodic modulation here needs to match the mechanical frequency, rather than the optical one. We plan to investigate in the future the reverse effect, where by externally driving the mechanical mirror, optical excitations in the USC system can be generated. On the experimental level, an implementation following the proposed schematics is within reach.

## Acknowledgements

We acknowledge useful discussions with Filippo Ferrari, Luca Gravina, Vincenzo Macrì, and Kilian Seibold.

**Author contributions**    F.M. and A.M. contributed equally.

**Funding information**    This work was supported by the Swiss National Science Foundation through Projects No. 200020_185015 and 200020_215172, and was conducted with the financial support of the EPFL Science Seed Fund 2021. M.S. acknowledges support from the EPFL Center for Quantum Science and Engineering postdoctoral fellowship. S.S. acknowledges support by the Army Research Office (ARO) through grant No. W911NF1910065.

## A  Analytical derivation of the steady state phonon number

Here we are interested in the steady state phonon population. We start by writing the effective Hamiltonian, and assuming $\omega_b \simeq \omega_d$ we have

$$
\begin{aligned}
\hat{H}_b(t) &= \omega_b \hat{b}^\dagger \hat{b} + \frac{g}{2} \mathcal{N}(t)(\hat{b} + \hat{b}^\dagger) = \omega_b \hat{b}^\dagger \hat{b} + \sum_j \frac{g}{2} \mathcal{N}_j(t)\left(e^{ij\omega_d t} + e^{-ij\omega_d t}\right)(\hat{b} + \hat{b}^\dagger) \\
&\simeq \omega_b \hat{b}^\dagger \hat{b} + \frac{g}{2} \mathcal{N}_1\left(e^{i\omega_d t} + e^{-i\omega_d t}\right)(\hat{b} + \hat{b}^\dagger) \\
&\simeq \omega_b \hat{b}^\dagger \hat{b} + \frac{g}{2} \mathcal{N}_1\left(\hat{b} e^{i\omega_d t} + \hat{b}^\dagger e^{-i\omega_d t}\right) = \hat{H}_b^{(1)}(t),
\end{aligned}
\tag{A.1}
$$

where $\mathcal{N}_1$ is the first Fourier component of $\mathcal{N}(t)$, and the approximation follows from being near the resonance condition and applying the rotating wave approximation. Finally, by passing in the frame rotating at the frequency $\omega_d$ through a time-dependent transformation $\hat{U}(t) = \exp(i\omega_d t \hat{b}^\dagger \hat{b})$, we obtain the time-independent Hamiltonian

$$
\hat{H}_b^{(1)} = \hat{U}(t)\hat{H}_b^{(1)}(t)\hat{U}^\dagger(t) - \omega_d \hat{b}^\dagger \hat{b} = -\Delta_1 \hat{b}^\dagger \hat{b} + \frac{g}{2} \mathcal{N}_1(\hat{b} + \hat{b}^\dagger),
\tag{A.2}
$$

where $\Delta_1 = \omega_d - \omega_s$.

As this set of transformations leaves the dissipative part of the system unchanged, we can now write the Lindblad master equation for the evolution of the reduced density matrix $\hat{\rho}_b$ of the phononic part as

$$
\dot{\rho} = -i\left[\hat{H}_b^{(1)}, \hat{\rho}_b\right] + (1 + n_{\text{th}})\gamma_b \mathcal{D}\left[\hat{b}\right]\hat{\rho}_b + n_{\text{th}}\gamma_b \mathcal{D}\left[\hat{b}^\dagger\right]\hat{\rho}_b + \gamma_{\text{D}} \mathcal{D}\left[\hat{b}^\dagger \hat{b}\right]\hat{\rho}_b,
\tag{A.3}
$$

so that the corresponding time evolution of the phonon population is governed by

$$
\frac{d}{dt}\left\langle \hat{b}^\dagger \hat{b}\right\rangle = \frac{d}{dt}\text{Tr}\left[\hat{b}^\dagger \hat{b}\hat{\rho}\right] = \text{Tr}\left[\hat{b}^\dagger \hat{b}\dot{\rho}\right]
\tag{A.4}
$$

Substituting Eq. (A.3) into Eq. (A.5), and expanding all the terms in normal-ordering, we obtain

$$
\frac{d}{dt}\left\langle \hat{b}^\dagger \hat{b}\right\rangle = -i\frac{g}{2}\mathcal{N}_1\left(\langle \hat{b}\rangle^* - \langle \hat{b}\rangle\right) - \gamma\left\langle \hat{b}^\dagger \hat{b}\right\rangle + n_{\text{th}}\gamma.
\tag{A.5}
$$

Similarly, the coherence evolve as

$$
\frac{d}{dt}\left\langle \hat{b}\right\rangle = i\Delta_1 - i\frac{g}{2}\mathcal{N}_1 - \frac{1}{2}(\gamma + \gamma_D)\left\langle \hat{b}\right\rangle,
\tag{A.6}
$$

At the steady state $d/dt(\langle \hat{b}^\dagger \hat{b}\rangle)_{\text{ss}} = d/dt(\langle \hat{b}\rangle)_{\text{ss}} = 0$, and we can now solve the linear system composed by the rhs of Eq. (A.5) and Eq. (A.6), whose solutions read

$$
\left\langle \hat{b}\right\rangle_{\text{ss}} = \frac{g\mathcal{N}_1}{2\Delta_1 + i(\gamma + \gamma_D)}
\tag{A.7}
$$

$$
\left\langle \hat{b}^\dagger \hat{b}\right\rangle_{\text{ss}} = \frac{\gamma + \gamma_d}{\gamma}\left|\left\langle \hat{b}\right\rangle_{\text{ss}}\right|^2 + n_{\text{th}}
\tag{A.8}
$$

## B  Open Dynamics using the full Hamiltonian

The full Liouvillian should contain additional terms, taking into account, e.g., particle loss also for USC part of the system, thus reading

$$
\dot{\rho} = \mathcal{L}_{\text{tot}}\hat{\rho} = \mathcal{L}\hat{\rho} + \gamma_a \mathcal{D}\left[\hat{X}^+(t)\right]\hat{\rho} + \gamma_\sigma \mathcal{D}\left[\hat{s}^+(t)\right]\hat{\rho}
\tag{B.1}
$$

where $\hat{\rho}$ is the density matrix of the system, $\gamma_{a,b,s}$ are the damping rates, while $\hat{X}^+(t)$ and $\hat{S}^+(t)$ are the positive frequency part of the dressed operators of the first and the second resonator, respectively [44]. These operators are obtained by expressing the field on the basis of the time-dependent eigenstates $|j(t)\rangle$ of the Hamiltonian (1), and by taking only the positive frequency part

$$
\begin{aligned}
\hat{X}_a^+(t) &= \sum_{j,k>j} \langle j(t)|\hat{a} + \hat{a}^\dagger|k(t)\rangle \, |j(t)\rangle\langle k(t)| \\
\hat{S}^+(t) &= \sum_{j,k>j} \langle j(t)|\hat{b} + \hat{b}^\dagger|k(t)\rangle \, |j(t)\rangle\langle k(t)| \,,
\end{aligned}
\tag{B.2}
$$

where $k > j$ means that the eigenvalue $E_k$ is larger than $E_j$. As, by construction, $\hat{X}_a^+$ and $\hat{S}^+(t)$ can only decrease the energy of the system, they can be safely neglected in the numerical simulations, as the USC part of the system adiabatically remains in the ground state.

It is worth noting that, contrary to the two resonators, the dissipation of the mirror is expressed in terms of the bare annihilation operator $\hat{b}$. Indeed: i) the optomechanical coupling we considered is very low; ii) We have no time dependence in the membrane dissipator since the virtual photons of the USC system act as an effective drive for the membrane, and drives must be introduced *after* the derivation of the dissipators. Failing to do that would lead to inconsistent results – for instance, a driven linear cavity would not emit any photon.

## C   Scheme for the experimental setup

As detailed in the main text and shown in Fig. 5, we propose the following circuit as an experimental platform to observe the phonon pump via excitation of the USC vacuum. A linear resonator (mode $\hat{a}$) is coupled to a transmon, that we model as a non-linear mode $\hat{c}$ with anharmonicity $\chi$. The "Rabi" Hamiltonian is thus

$$
\hat{H}_R = \omega_a \hat{a}^\dagger \hat{a} + [\omega_\sigma + \Delta_\omega \sin(\omega_d t)]\hat{c}^\dagger \hat{c} + \chi \hat{c}^\dagger \hat{c}^\dagger \hat{c}\hat{c} + \lambda(t)\left(\hat{a} + \hat{a}^\dagger\right)\left(\hat{c} + \hat{c}^\dagger\right)
\tag{C.1}
$$

Notice that, in this configuration, the coupling between the resonator and the transmon depends on their relative frequency, and thus we have introduced the coupling

$$
\lambda(t) = \sqrt{\omega_a\left[\omega_\sigma + \Delta_\omega \sin(\omega_d t)\right]}\frac{C_c}{2\sqrt{(C_a + C_c)(C_\sigma + C_c)}} = \lambda_0\sqrt{1 + \Delta_\omega \sin(\omega_d t)/\omega_\sigma}\,,
\tag{C.2}
$$

where $C_a$ is the capacity of the linear resonator, $C_\sigma$ that of the transmon, and $C_c$ is the capacitive coupling between the two, and

$$
\lambda_0 = \sqrt{\omega_a \omega_\sigma}\frac{C_c}{2\sqrt{(C_a + C_c)(C_\sigma + C_c)}}
\tag{C.3}
$$

is the coupling when the modulation is turned off. The optomechanical coupling maintains its form and reads

$$
\hat{H}_{\text{opt}} = \omega_b \hat{b}^\dagger \hat{b} + \frac{g}{2}(\hat{a} + \hat{a}^\dagger)^2(\hat{b}^\dagger + \hat{b}).
\tag{C.4}
$$

## D   Derivation of the effective mirror Hamiltonian

Before performing the factorization and the adiabatic approximation, it is useful to remove the static mirror displacement term, which arises from the optomechanical interaction $(\hat{a}+\hat{a}^\dagger)^2(\hat{b}+\hat{b}^\dagger)$.

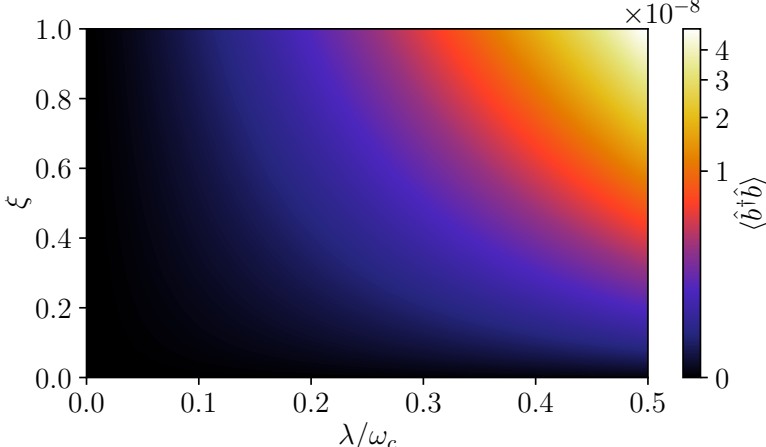

Figure 6: Influence of the counter-rotating terms and the cavity-qubit coupling on the generation of phonons. The number of phonons is taken after 100 cycles of the closed dynamics, starting from the zero phonons state. As can be seen, both the counter-rotating terms and the large coupling are required to achieve this effect. The used parameters are the same as in Fig. 2 of the main text.

Indeed, by expanding this term, we have

$$\left(\hat{a} + \hat{a}^\dagger\right)^2 \left(\hat{b} + \hat{b}^\dagger\right) = \left(2\hat{a}^\dagger\hat{a} + \hat{a}^2 + \hat{a}^{\dagger 2} + 1\right)\left(\hat{b} + \hat{b}^\dagger\right) = \left(2\hat{a}^\dagger\hat{a} + \hat{a}^2 + \hat{a}^{\dagger 2}\right)\left(\hat{b} + \hat{b}^\dagger\right) + \hat{b} + \hat{b}^\dagger,$$
(D.1)

and the last term causes a static displacement of the mirror, which can be seen as an energy re-normalization after a transformation.

Let's take the displacement operator $\hat{\mathcal{D}}(\beta) = \exp(\beta\hat{b}^\dagger - \beta^*\hat{b})$, which transforms the operators as $\hat{\mathcal{D}}(\beta)\hat{b}\hat{\mathcal{D}}^\dagger(\beta) = \hat{b} - \beta$. By rotating the total Hamiltonian in Eq. (**??**)eq: generic total hamiltonian of the main text, we have that only $\hat{H}_{\mathbf{opt}}$ is affected, obtaining

$$\begin{aligned}
\hat{H}'_{\mathbf{opt}} = \hat{\mathcal{D}}(\beta)\hat{H}_{\mathbf{opt}}\hat{\mathcal{D}}^\dagger(\beta) = {} & \omega_b\hat{b}^\dagger\hat{b} - \omega_b\beta\hat{b}^\dagger - \omega_b\beta^*\hat{b} + \omega_b|\beta|^2 \\
& + \frac{g}{2}\left(2\hat{a}^\dagger\hat{a} + \hat{a}^2 + \hat{a}^{\dagger 2}\right)\left(\hat{b} + \hat{b}^\dagger\right) - \frac{g}{2}\left(2\hat{a}^\dagger\hat{a} + \hat{a}^2 + \hat{a}^{\dagger 2}\right)(\beta + \beta^*) \\
& + \frac{g}{2}\left(\hat{b} + \hat{b}^\dagger\right) - \frac{g}{2}(\beta + \beta^*) \, .
\end{aligned}$$
(D.2)

By choosing $\beta = g/2\omega_b$, and neglecting terms in $g^2$, we get

$$\hat{H}'_{\mathbf{opt}} = \omega_b\hat{b}^\dagger\hat{b} + \frac{g}{2}\left(2\hat{a}^\dagger\hat{a} + \hat{a}^2 + \hat{a}^{\dagger 2}\right)\left(\hat{b} + \hat{b}^\dagger\right) \, .$$
(D.3)

The terms of the order of $g^2$ include a constant energy shift $-g^2/(4\omega_b)$, a frequency shift of the resonator $-g^2/(2\omega_b)\hat{a}^\dagger\hat{a}$, and a two photon drive $-g^2/(4\omega_b)(\hat{a}^2 + \hat{a}^{\dagger 2})$. Neglecting all these terms does not affect the purposes of this work, and their contribution is minimal due to the small optomechanical coupling we have chosen ($g = 2\pi \times 15$ Hz).

Terms displacing the mirror thus generate a new equilibrium position, around which the mirror vibrates. By definition, dissipation and drives act around this rest position of the mirror. Failing to consider these constant shifts emerging from the various coupling results in unphysical effects. For instance, one could generate infinite energy as the mirror could dissipate in a bath (thus releasing energy) but still be driven by constant terms.

# E  Influence of the counter-rotating term and effect of the ground-state fluctuations

To show that the predicted effect is solely due to the modulation of the vacuum through USC, here we consider an artificial model where we independently tune the rotating and the counter-rotating terms. Namely, we set

$$\hat{H}_{\mathrm{R}} = \omega_a \hat{a}^\dagger \hat{a} + \omega_\sigma \hat{\sigma}_+ \hat{\sigma}_- + \lambda(\hat{a}\hat{\sigma}_+ \hat{a}^\dagger \hat{\sigma}_-) + \xi\lambda(\hat{a}\hat{\sigma}_- + \hat{a}^\dagger \hat{\sigma}_+), \tag{E.1}$$

with $\xi \in [0,1]$. When $\xi = 1$, this corresponds to the full Rabi model considered in the main text. The approximation $\xi = 0$ is known as the Jaynes-Cumming model, and it is valid only in the limit $\lambda \ll \omega_a, \omega_\sigma$. All other terms in the Hamiltonian are kept as in the main text.

In Fig. 6, we show that $\xi \neq 0$ is a fundamental factor to observe the wanted effect. Despite the qubit modulation, in the absence of counter-rotating terms, the phononic mode never oscillates. Furthermore, larger values of $\lambda$ results in higher visibility of the effect.

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
