# Peer review of "Phonon Pumping by Modulating the Ultrastrong Vacuum"

_SciPost Physics_

## Round 1 · Referee Report · Anonymous (Referee 1) · 2024-6-5

Strengths

1 - proposal of a novel effect

2- The proposed effect is quite straightforward and there is no reason to doubt the validity of the predictions. The paper contains an analytical approximation to the phonon number that is validated by full numerical simulations.

3- An experimentally realistic implementation of the predicted effect with superconducting circuits is proposed.

4- The paper is clearly written and easy to follow, even for non-experts in the field. The relevant literature is well cited.

5- The derivations of the model and results are clearly presented in the appendices.

Weaknesses

1- the predicted effect is not very large in the proposed superconducting circuit system. Since this is the theoretical and therefore typically idealised estimate, there is some doubt whether this effect is experimentally observable.

Report

By presenting a novel effect by considering a optomechanical system in the strongly coupling regime, the manuscript meets on of the expectations for publication in SciPost.

I therefore recommend the publication of the manuscript.

Requested changes

1- line 264, there is a reference to an equation that is missing

2- line 200-201: in the outlook, it is said that the authors plan to investigate to excited optical excitations by driving the mirrors. To me, it seems that this outlook faces the problem of generating mechanical vibrations that are fast enough to be resonant with the optical modes, a condition that seems not to be facilitated by working in the USC regime. The authors could reconsider whether this outlook is realistic and refer to systems where this resonance condition could be met.

Recommendation

Publish (meets expectations and criteria for this Journal)

---

## Round 1 · Referee Report · Anonymous (Referee 2) · 2024-6-7

Strengths

This is a very original and interesting study, which provides insights into an effect generated by modulating the ultrastrong vacuum limit of cavity QED. I strongly recommend it for publication in SciPost, after the authors clarify the issues listed in the report below.

Weaknesses

The manuscript is reasonably strong and I do not see major weaknesses. Some minor weak points (relatively easy to fix) are listed below.

Report

Referee Report for the manuscript titled:

Phonon Pumping by Modulating the Ultrastrong Vacuum.

This is a very original and interesting study, which provides insights into an effect generated by modulating the ultrastrong vacuum limit of cavity QED. I strongly recommend it for publication in SciPost, after the authors clarify the following issues:

1. Typically, circuit QED systems are affected by pure dephasing. Why is pure dephasing missing in the dissipative terms?

2. What happens if we use a different qubit (e.g., a Flux qubit)? Would there be any advantage to using a different configuration?

3. The effect is visible according to the authors' simulation and estimation of experimentally achievable parameters. What would make the effect even more visible to allow for an even easier experimental implementation?

4. It would be preferable if the authors consider citing additional references on related effects obtained by modulating the quantum vacuum.

In summary, this is an excellent work, which could be published in SciPost after the authors consider the suggested improvements listed above.

Requested changes

Please see the report, listing requested changes.

Recommendation

Publish (surpasses expectations and criteria for this Journal; among top 10%)

---

## Editorial Decision

resubmitted